# Atomic Order and Submicrostructure in Iron Alloys at Megaplastic Deformation

**Valery Shabashov \*, Victor Sagaradze, Kirill Kozlov and Yury Ustyugov**

Mikheev Institute of Metal Physics, Ural Branch, Russian Academy of Sciences, 620108 Ekaterinburg, Russia; vsagaradze@imp.uran.ru (V.S.); kozlov@imp.uran.ru (K.K.); ustyugov@imp.uran.ru (Y.U.)

\* Correspondence: shabashov@imp.uran.ru; Tel./Fax: +7-343-3745244

**Abstract:** The subject of the present review consists of summing up our previous results on the study of the relaxation of structure along the way (i) of atomic redistribution—in the form of short-range clustering in binary iron alloys—induced by megaplastic deformation (i.e., of super large value), and (ii) of the dissolution and precipitation of disperse nitrides and carbides in steels and intermetallics in ageing alloys. Within the capacity of the main method of executing megaplastic deformation, along with the practically important milling in ball mills and friction-providing external action, we employed high pressure torsion (HPT) in Bridgman anvils, which permitted the control of the degree, rate, and temperature of deformation action. At the local level of two nearest neighbors (one or two coordination shells in relation to an iron atom) we studied atomic mass transfer, stipulated by generation of a large number of point defects of deformation origin, and conducted a comparison with a case of irradiation by high-energy electrons. We established a change in the direction of phase transformations, as well as anomalous acceleration of the ordering and precipitation of disperse phases upon altering the temperature ($T < 0.3 T_{melt}$) and rate of deformation (from $2 \times 10^{-2}$ to $8 \times 10^{-2}$ s$^{-1}$). We also demonstrated the possibility of regulating the ultra-fine-grained structure with solid–solution strengthening and dispersion hardening.

**Keywords:** iron alloys; steels; deformation; mechanosynthesis; irradiation; ordering; point defects; Mössbauer spectroscopy

## 1. Introduction

In modern materials science, the scientific direction of deformation-induced nanostructuring and anomalous phase transformations, induced by super-large (mega) plastic deformations [1–7], has acquired active development. The topicality of this direction is stipulated by the practical importance of inventions of dispersion-hardened steels, alloys, and nanocomposites [8–12].

Dissipation of the energy of large plastic deformation is accompanied by anomalous structure–phase transformations and cardinally changes the properties of metals and alloys. Such phenomena are observed as mechanosynthesis (via mechanical alloying), low-temperature lattice diffusion, etc. [3–6]. In the paper ref. [7], the term (of) megaplastic deformation was introduced, and an attempt was made to formulate a unified concept of the process of super-high plastic deformation of solid bodies. The physical meaning of this notion consists of the fact that that because of the decline in the mobility of dislocations and a decrease in the ability of relaxation of the stored elastic energy along the way of structure fragmentation, the following processes are possible; (i) dynamic recrystallization, (ii) stress-assisted diffusion, (iii) thermally activated diffusion, and (iv) amorphization at temperatures $T/T_{melt} < 0.3$. The border of realization of the transition from the macro- to the mega-level is determined by the switching-on of the mechanisms for changing the channels of relaxation along the way of atomic mass transfer, recrystallization, and formation of nanostructures. The appearance of the works refs. [5,9–23] on the induced dissolution of disperse

particles, formation of supersaturated solid solutions, and dynamical aging, namely, accelerated ordering and decomposition with the formation of secondary phases that strengthen a metallic matrix, has enabled a development in studies on deformation-induced nanostructuring and mechanosynthesis. So, the effect of the conditions of mechanical activation on the mechanism of transformations and on the final modified structure of steels and alloys has been elucidated.

In the present review the main attention has been paid to the results of our original investigations over the last five years on the phase transformations in iron alloys and steels induced by large plastic deformations. The authors of these works were initiators of the first studies on anomalous low-temperature phase transformations [24–31]. Analysis of the effect of the temperature and rate of large plastic deformation on the mechanism of phase transformations, as well as a comparison with the case of irradiation action, has become a specific feature of these works [13–20]. An attempt to compare the cases of deformation action and irradiation by high-energy elementary particles is explained by the generation in both cases of mobile point defects, namely, vacancies and interstitial atoms. The participation of point defects was analyzed at the initial stages of the (i) phase transformations in the form of the processes of the formation and destruction of short-range clustering in binary Fe-(Mn, Cr, Ni) iron alloys [18–20], of (ii) the formation and decomposition of the solid solutions of ferritic and austenitic Fe-(Mn, Cr, Ni)-N and Fe-Ni-C steels, supersaturated by nitrogen and carbon [10–12,19,21–23], as well as of (iii) the dissolution and precipitation of intermetallic particles in ageing Fe-Ni-(Ti, Al, Si, Zr) alloys [13–17].

To execute large plastic deformation and to control the conditions of the experiment, along with practically important external actions, such as milling in a ball mill (BM) [9–11], frictioning at friction treatment [21,32,33], and reducing at rolling [15], the high pressure torsion (HPT) method (of pressure torsion) in Bridgman anvils [12–21,23] was used. The method of HPT permits (i) the creation of high degrees of true deformation ($\varepsilon \sim 9$) without the failure of samples and (ii) the control of the degree, rate, and temperature of deformation. The degree of the true deformation $\varepsilon$ of a disc-shaped sample at HPT (at a radius of $r$ from the center of the sample) was determined (taking account of the sample's upsetting) via the simplified formula [34]:

$$\varepsilon = \ln(h_0/h) + \ln(2\pi nr/h) \tag{1}$$

where $n$ is the number of revolutions of a rotating Bridgman anvil, $h_0$ and $h$ are the initial and the final thickness of the sample. The rate of deformation was varied from $2.4 \times 10^{-2}$ s$^{-1}$ (0.3 rpm) to $8 \times 10^{-2}$ s$^{-1}$ (1 rpm). The temperature of deformation was measured from 80 to 573 K. The scheme of the HPT experiment is given in Figure 1. To obtain a non-ordered state of the materials, the annealing was performed at a high temperature. This state of the material is further referred to as as-received. Apart from this, the effect of the deformation on the preliminarily aged steels and alloys was studied. As the main method of structural investigations, transmission Mössbauer spectroscopy was used which makes it possible to analyze the structure and atomic redistribution in iron alloys at the local level of the nearest neighborhoods (1–2 coordination shells) of the resonant $^{57}$Fe.

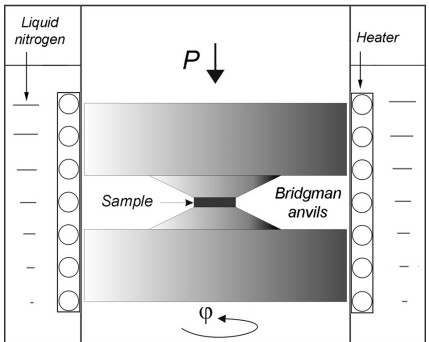

**Figure 1.** A schematic of experiment on deformation of a sample in Bridgman anvils, in the temperature range 80–573 K.

## 2. Dynamic Short-Range Clustering in Supersaturated Iron Alloys upon Megaplastic Deformation

### 2.1. Acceleration of Short-Range Clustering in the Binary Alloys Fe–Mn(Cr,Ni) at "Warm" Megaplastic Deformation

The short-range clustering that forms aggregations of atoms and the different nature in the crystal lattice of alloys can represent the initial stage of phase transformations. In the works [18–20] one can find an authorized consideration of the results of the effect of large plastic HPT and BM deformations on short-range clustering in binary Fe-Cr, Fe-Mn, and Fe-Ni alloys, as well as compared these results with the case of irradiation by high-energy (5.5 MeV) electrons.

The control over distribution of atoms in the binary alloys $Fe_{100-x}Cr_x$ ($x$, at % = 12, 13.2, 21.9) and $Fe_{100-x}Mn_x$ ($x$, at % = 4.1, 6.8, 9, 9.9) with BCC crystal lattice was carried out via separation of subspectra—the sextets $S$ ($N_1$, $N_2$) that correspond to the non-equivalent enveloping of iron with the impurity atoms (Cr, Mn) with $N_1$ in the first and $N_2$ species in the second coordination shells (CS) [35,36], see Figure 2. Estimation of the distribution was carried out via calculating the occupancies of non-equivalent surroundings of iron, $W$ ($N_1$, $N_2$), which were proportional to the intensity of the sextets $S(N_1, N_2)$. The degree of ordering was determined based on the mean effective concentration of impurity in the two CS nearest to Fe, $\bar{c}$, and by the Warren–Cowley parameter $\alpha_{1,2}$ [18,20,35,36]. In the case of statistically uniform (binominal) distribution of atoms over the sites of lattice, the value of $\bar{c}$ is equal to the atomic concentration of impurity, $x$, in the alloy, and of $\alpha_{1,2} = 0$. On the establishment in the alloy of short-range clustering of the type of short-range order (SRO), i.e., with tendency to neighborhood of atoms of different nature, we have $\bar{c} > x$ and $\alpha_{1,2} < 0$, and at short-range separation (SRS), i.e., with tendency to neighborhood of atoms of the same nature, we have $\bar{c} < x$ and $\alpha_{1,2} > 0$.

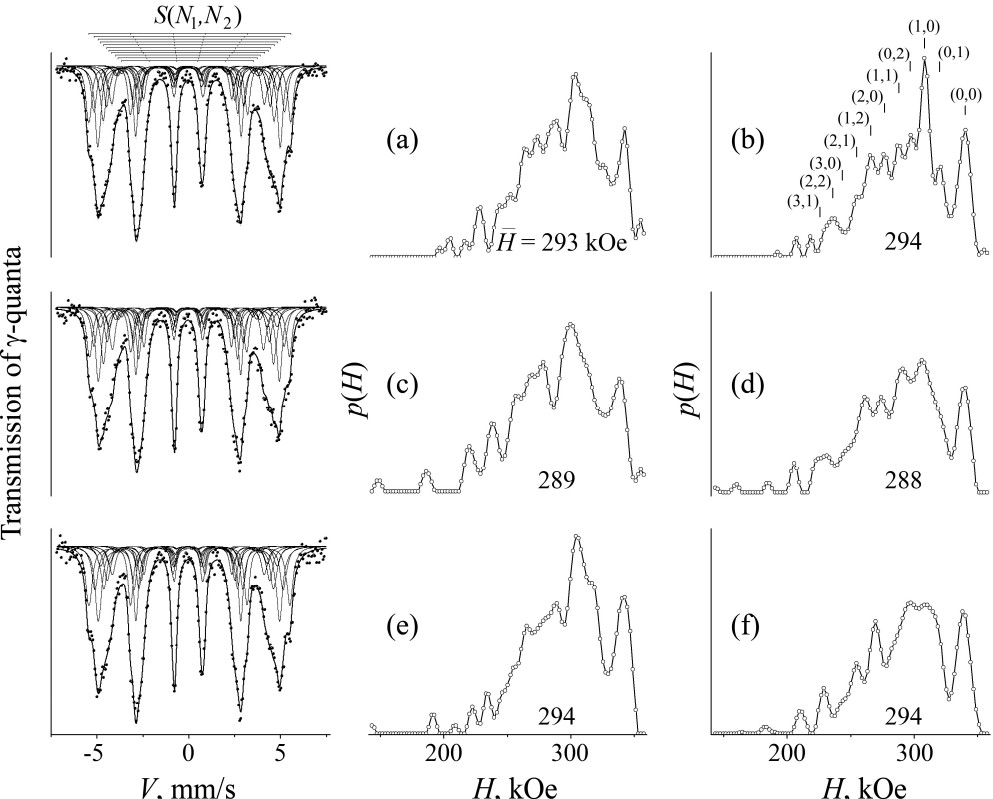

**Figure 2.** Spectra, distributions $p(H)$, and $\overline{H}$ for the alloy $Fe_{86.8}Cr_{13.2}$. Treatment (state): (**a**) annealed at 1073 K (as-received); (**b**) annealed at 773 K, 50 h (aged); (**c**) as-received and high pressure torsion (HPT) at 80 K; (**d**) aged and HPT at 80 K; (**e**) as-received and HPT at 573 K; (**f**) aged and HPT at 573 K. The $p(H)$ and $\overline{H}$ are shown only for the annealed alloy. (b) (shown) positions of the hyperfine fields corresponding to the surroundings ($N_1$, $N_2$).

The alloys $Fe_{100-x}Cr_x$ and $Fe_{100-x}Mn_x$ under investigation after annealing or irradiation by electrons reveal the formation of short-range clustering according to the type of the SRS between Cr and Mn in $\alpha$-Fe [35,36]. At the same time, impurity atoms tend to mix in the thermodynamically favorable surrounding of atoms of their same nature. With the example of the alloy $Fe_{86.8}Cr_{13.2}$ one can see the growth of the partial contribution of the sextets $S(N)$, $N = N_1 + N_2$ that correspond to the surroundings of iron atoms of low impurity content, Figure 2a,b. In the distributions of hyperfine magnetic fields, $p(H)$, one can also observe in high-magnitude fields the increase of the intensity of the mean hyperfine magnetic field $\overline{H}$. In the work [18] the authors showed that in conditions of «cold» HPT deformation (of 80 and 298 K) of the alloy $Fe_{86.8}Cr_{13.2}$ obtained in the as-received state after annealing at 1073 K for 4 h and preliminarily aged at 773 K for 50 h, there takes place a decrease in the partial contribution of the sextets corresponding to the surroundings of iron without, or with small amounts of the atoms of Cr ($N \leq 0.1$), Figure 2c,d, i.e., the effective concentration $\overline{c}(Cr)$ exhibits an increase from 11.8 to 12.5 at %,—the value close to the alloy chemical composition ($x = 13.2$ at %), Figure 3a. This means that the distribution of impurities close to that which is characteristic of a disordered state becomes the result of «cold» deformation. «Warm» (473–573 K) deformation is accompanied by the opposite effect, namely, by the growth of the partial contribution of the sextets $S$ at the expense of the surroundings of iron depleted of the impurity Cr and by the decrease of $\overline{c}(Cr)$ down to 10.8 at %, see Figures 2e,f and 3a. Note that the thermal annealing at 773 K for 50 h, decreases the $\overline{c}(Cr)$ down to 11.6 at %, i.e., one observes here an achievement of a degree of separation lower than that obtained in the HPT process at 573 K for 10 min (3 revs at 0.3 rpm).

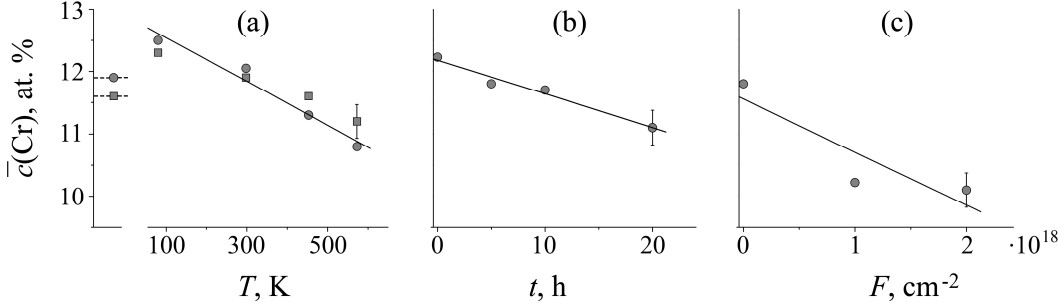

**Figure 3.** (**a**,**b**) The dependence (and (**c**) a tendency) of $\overline{c}(Cr)$ in the matrix of the alloys (**a**), (**b**) $Fe_{86.8}Cr_{13.2}$, and (**c**) $Fe_{88}Cr_{12}$ on the temperature of HPT, duration of ball milling (BM), and fluence of electrons at 423 K. Round symbols correspond to the values for the state "annealed at 1073 K (as-received)"; square symbols, for the state "annealed at 773 K, 50 h (aged)".

Such processes of deformation-induced atomic redistribution are observed in the alloy $Fe_{100-x}Mn_x$ ($x = 4.1, 6.8, 9, 9.9$) [20]. In Figure 4 one can see Mössbauer spectra for the alloy $Fe_{93.2}Mn_{6.8}$ after its corresponding treatments. Changes in the mean effective concentration are presented in Figure 5. After HPT at 80 and 298 K, in the as-received and aged alloys disordering occurs, i.e., a decrease of the degree of separation, while after HPT at 473 and 573 K, an additional separation over Mn is observed.

In the invar FCC alloy $Fe_{64.9}Ni_{35.1}$ analogous results were obtained, namely, a destruction of the homogeneous short-range clustering after «cold» (of 80 and 298 K) HPT deformation, and, on the contrary, the growth of the degree of short-range clustering after HPT at 473 and 573 K [19,22]. After annealing at 873 K for 1 h, in the alloy, homogeneous short-range clustering is formed according to the SRO type with $\alpha < 0$, which manifests itself in the increase of the partial contribution into $p(H)$ of high-magnitude hyperfine magnetic fields $H_h$ and in the growth of $\overline{H}$ from 235 to 246 kOe, Figure 6a. Subsequent HPT deformation ($\varepsilon = 4.2$, $n = 1$ rev.) at 80 and 298 K returns the $p(H)$ and $\overline{H}$ parameters to those characteristic of the initial (as-received) alloy water-quenched from 1073 K. As a result of the HPT ($\varepsilon = 7$, $n = 10$ revs) at 473 and 573 K, changes analogous to those obtained after irradiation by high-energy electrons (5.5 MeV) in the same range of temperatures become a reality, Figure 6b,c. In the spectrum both the component $H_h$ appears—with the value of field in the vicinity of 285–290 kOe,

corresponding to enrichment in nickel of the surroundings of resonant iron—as well as the central peak $H_l$ of the paramagnetic austenite depleted of nickel. The character of the spectra and $p(H)$ distributions reflects the short-range clustering that is usually connected with both the existence of the immiscibility ranges and the ordering of the type characteristic of FeNi [37–39].

Thus, from the results presented above it follows that a change in the temperature of large plastic deformation in the as-received and aged Fe-(Cr, Mn, Ni) alloys leads to the changeover of the direction of atomic redistribution from the destruction of the atomic order upon «cold» deformation to the accelerated short-range clustering at «warm» deformation.

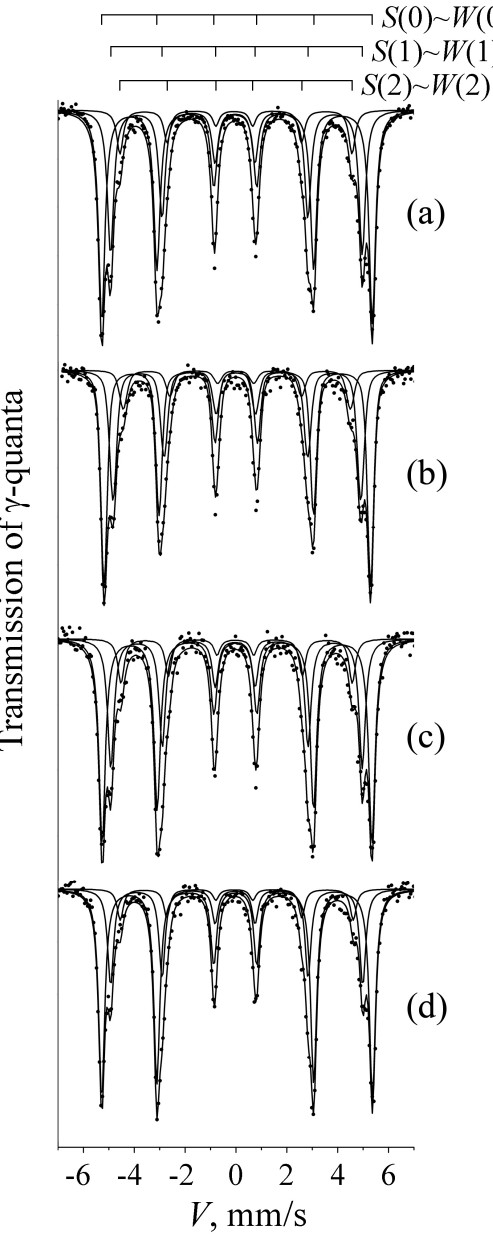

**Figure 4.** Spectra for the alloy $Fe_{93.2}Mn_{6.8}$. Treatment (state): (**a**) annealed at 1073 K (as-received), (**b**) annealed at 773 K, 1 h (aged); (**c**) as-received and HPT at 80 K; (**d**) as-received and HPT at 573 K.

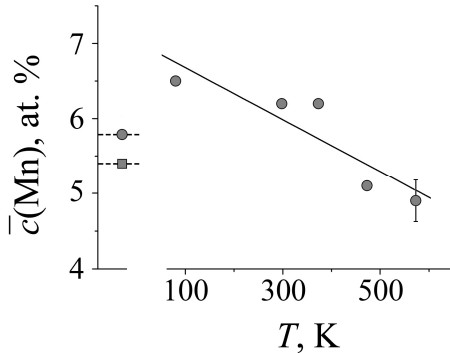

**Figure 5.** The dependence of $\bar{c}$(Mn) in the matrix of the as-received alloy Fe$_{93.2}$Mn$_{6.8}$ (i.e., annealed at 1073 K) after its HPT at different temperatures. Square symbols point out the concentration $\bar{c}$(Mn) composition of the alloy aged at 773 K for 1 h.

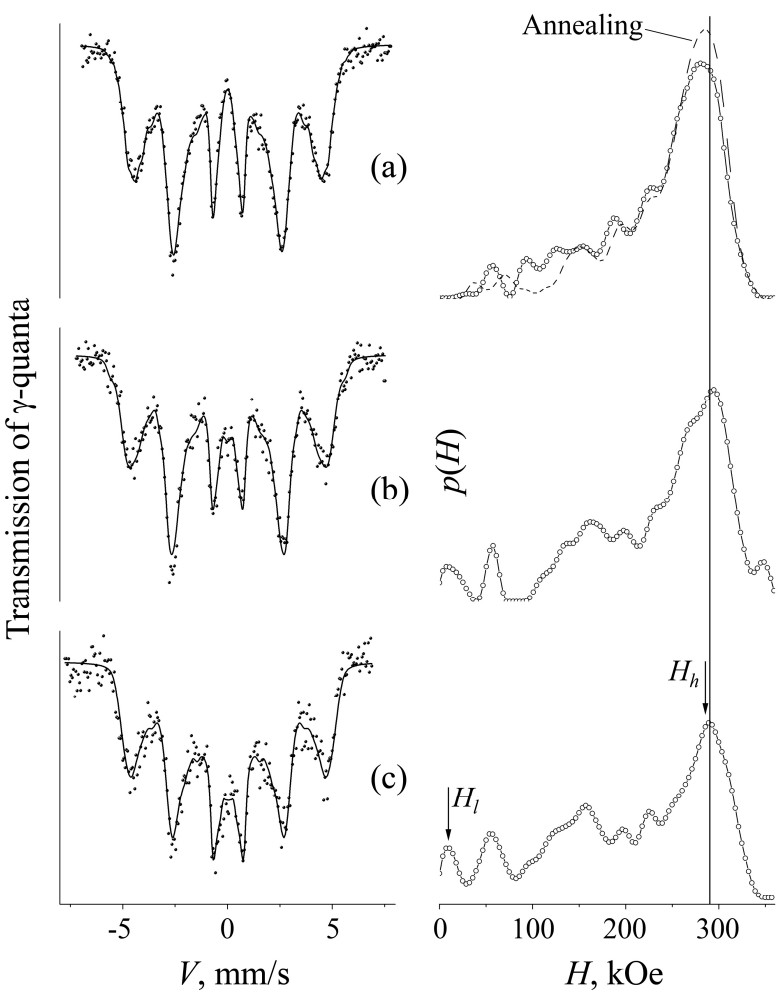

**Figure 6.** Spectra and distributions $p(H)$ for the alloy Fe$_{64.9}$Ni$_{35.1}$. Treatment (state): (**a**) water-quenched from 1323 K (as-received), dashed line indicates $p(H)$ for the state "annealed at 873 K for 1h" (aged); (**b**) as-received, aged, and HPT treated ($\varepsilon$ = 7, $n$ = 10) at 573 K; (**c**) as-received and subsequently irradiated by electrons of 5.5 MeV with (fluence of) $F = 5 \times 10^{18}$ cm$^{-2}$ at 573 K. In the distributions $p(H)$ we pointed out the peaks $H_l$ and $H_h$, that correspond to the $\gamma$ matrix depleted of, and enriched in, nickel.

### 2.2. The Mechanism and Kinetics of Deformation-Induced Ordering in Iron Alloys. A Comparison with the Case of Irradiation by High-Energy Electrons

The atomic redistribution revealed in the Fe-Cr, Fe-Mn, and Fe-Ni alloys after their HPT treatment at 473–573 K is qualitatively similar to the redistribution resulting from the thermal anneals and irradiation by high-energy electrons [35–39]. Alternations in the degree of ordering mean a shift of the relation between the processes of destruction and formation of order towards enhancement of order at rising temperature of deformation. One can suppose that such a shift is connected with the increase in the rate of short-range clustering, which migrating defects provide at rising temperature, while the rate of destruction of short-range clustering by moving dislocations is weakly dependent on the temperature. This conclusion is confirmed by an opposite change in the partial contributions of $S(N)$ when lowering the deformation temperature from 573 K down to 80 K. At 80 K one can expect a sufficient decrease in the rate of short-range clustering at the expense of migration of defects at a virtually unchanged rate of the destruction of short-range clustering by moving dislocations. Indeed, after HPT at 80 K a destruction of the initial short-range clustering is observed both of as-received and aged alloys, Figures 2, 3 and 5. Approaching an effective concentration of chromium, manganese, and nickel to the chemical composition $x$ after «cold» deformation means that the initial quenched condition of the alloys is not completely disordered.

Atomic separation in the Fe-Cr and Fe-Mn alloys in the course of their BM milling, Figure 3b, also can be related to the strengthening of the role of mobile non-equilibrium defects at high local temperatures in the regions of mutual collision between the balls and particulates. Correspondingly, the duration of the action of an enhanced temperature is proportional to the number of impacts or total time of milling. Therefore, the degree of separation is increased when increasing the duration of milling. Undoubtedly, the mechanisms of deformation at HPT and BM have their own specific characteristic features. Nonetheless, the main result of these external actions is the acceleration of the atomic mass transfer considerably surpassing the atomic redistribution on annealing under stationary laboratory conditions.

When analyzing the mechanism and kinetics of the atomic mass transfer (AMT) induced by large-magnitude deformation, one should keep in mind its complex character. In most cases, this makes one consider only the main contribution to the AMT. Separation in conditions of HPT and BM turns out to be considerably more efficient in comparison with that in the conditions of thermal effect. The parameter Cowley $\alpha_{1,2} = 1 - \frac{<c^*>}{c}$ of the alloy $Fe_{86.8}Cr_{13.2}$ after the alloy's subjection to HPT at 453 K (over 10 min) amounts to 0.14. At the same time, irradiation by electrons at 423 K with (a fluence of) $F = 2 \times 10^{18}$ cm$^{-2}$ gives the value $\alpha_{1,2} \approx 0.1$. This is, that the degree of separation accelerated by deformation is close to the degree of separation obtained as a result of electron irradiation [35], Figure 3. A similar regularity in the effect of HPT temperature on the acceleration of separation is observed in supersaturated Fe-Mn alloys. Thus, in the alloy $Fe_{93.2}Mn_{6.8}$ after HPT at 298 K and subsequent annealing at 773 K (8 h), the change of the effective concentration is sufficiently lower than in the case after HPT at 573 K [20]. The origin of accelerated separation at «warm» deformation, as in the case of radiation influence, is explained as being due to a permanent generation of mobile point defects in the course of the process of external impact. Being the item of principle significance, the continuity of the deformation action for atomic redistribution and lattice diffusion at relatively moderate temperatures was underlined in ref. [6] as a condition of structure saturation with the interstitial atoms and vacancies. The possibility of destruction of short-range clustering, as well as the fact of attainment of mechanical alloying at cryogenic temperatures, is explained by the generation and movement of interstitial atoms in the stress field of the moving dislocations [4–6,16,29,40–42]. Interstitial atoms, which are capable of movement at low temperatures, participate both in the processes of disordering, moving in the stress field of moving dislocations, as well as in the processes of ordering. Short-range clustering is stipulated by the limited mixability between iron and impurity in the range of the compositions and temperatures under investigation. In the range of the temperatures higher than room temperature, the prevailing mechanism that determines the efficiency of deformation-induced ordering is allowed to be

the one which is crucially dependent on the value of the momentary concentration of vacancies [18–20]. According to XRD data, after equal channel angular pressing (ECAP), the concentration of vacancies in the structure increases by several orders of magnitude [43,44]. The large concentration of vacancies at decreasing temperature leads to the growth of the possibility of the formation of mobile vacancy complexes of limited size. With rising complexity of a defect its diffusion efficiency and participation in the process of short-range clustering is expected to undergo a sufficient decrease.

## 3. The Formation of Submicrostructure in High-Nitrogen Fe-Cr-N and High-Carbon Fe-Ni-C Steels upon Megaplastic Deformation

The possibilities to control and govern atomic redistribution by means of large plastic deformation have found further realization in the works on deformation-induced dissolution and precipitation of disperse interstitial phases in austenitic and ferritic steels [10–12,21–23,32,33].

### 3.1. The Effect of Deformation Temperature on the Dissolution and Precipitation of Nitrides and Carbides in Steels

To investigate the effect of the temperature of the megaplastic deformation in Bridgman anvils on the structure of nitrogen- and carbon-supersaturated steels $Fe_{71.2}Cr_{22.7}Mn_{1.3}N_{4.8}$, at %, [22,23] and $Fe_{64.7}Ni_{33.6}C_{1.7}$, at %, [19,22], we employed the methods of Mössbauer spectroscopy and electron microscopy.

The high-nitrogen austenitic steel $Fe_{71.2}Cr_{22.7}Mn_{1.3}N_{4.8}$ after its quenching at 1453 K in water and subsequent aging at 923 K for 2.5 h decomposes with the formation of a perlite-like structure containing alternating 50 μm-thick interfacial layers of the nitride $Cr_2N$ and ferrite, see Figure 7a. The control over the precipitation and dissolution of such sufficiently coarse nitrides of chromium upon deformation was performed by measuring the effective chromium concentration in a solid solution of ferritic BCC $\alpha$ phase. The concentration $\bar{c}(Cr)$ was determined by the partial contribution of subspectra of $S(N)$ proportional to $W(N)$—the possibility of occupation by $N$ chromium atoms, of the nearest two coordination shells (CS) of iron [18,45,46]. The Mössbauer spectra and values of $\bar{c}(Cr)$ before and after HPT are shown in Figures 8 and 9. HPT at 80 and 293 K increases $\bar{c}(Cr)$ in value from 18.5 to 21.9 at %, which, on taking into account the stoichiometry of $Cr_2N$, corresponds to a growth of 1 at % N of the nitrogen concentration in the ferritic matrix. The opposite result, namely, the absence of growth or even the decrease of $\bar{c}(Cr)$ in value in the matrix from 18.5 to 18 at %, is observed after HPT at 573 K (Figure 9). The change of composition is confirmed by the magnitudes of the mean hyperfine magnetic field $\bar{H}$ having linear dependence on $\bar{c}(Cr)$ [47] (Figure 8). Therefore, the increase of deformation temperature to 573 K intensifies the decomposition of the BCC solid solution preliminarily aged at 923 K, which is reliably confirmed by the Mössbauer method data. The electron microscopy data testify to an intense fragmentation of the nitride lamella and ferrite grains as a result of megaplastic deformation at 298 and 573 K (Figure 7b,c). In the XRD pattern taken on the sample of a steel HPT deformed at 573 K, apart from $\alpha$-phase, one can observe reflections from the nitrides $Cr_2N$. By the morphology, the small particles of nitrides do not differ from those obtained as a result of deformation at 298 K, but the quantity of the nitride particles increases.

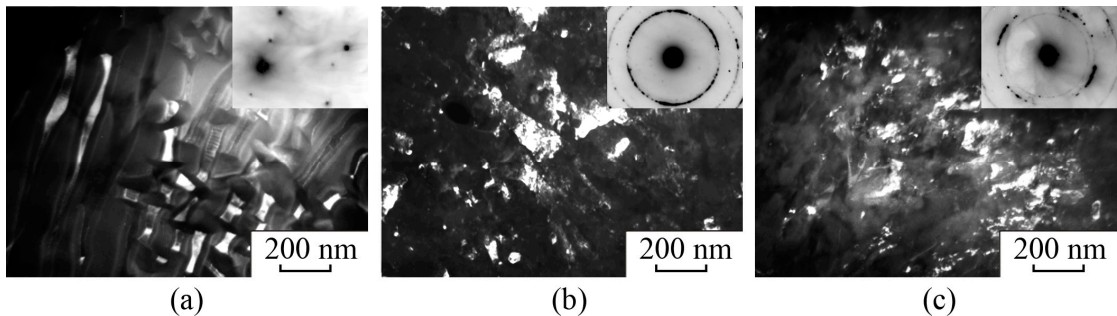

**Figure 7.** The structure of the steel $Fe_{71.2}Cr_{22.7}Mn_{1.3}N_{4.8}$. Treatment (state): (**a**) annealed at 923 K for 2.5 h (aged); (**b**) aged and HPT treated at 298 K; (**c**) aged and HPT treated at 573 K. (**a**) dark-field image taken in the reflection $(00\bar{2})_{CrN}$ with XRD pattern, zone axis $[\bar{1}37]_\alpha$ and $[101]_{Cr2N}$; (**b**) dark-field image taken in the reflections $(002)_{Cr2N} + (110)_\alpha$ with XRD pattern (marked the rings from $Cr_2N$ and $\alpha$); (**c**) dark-field image taken in the reflection $(2\bar{1}0)_{Cr2N}$ with XRD pattern (marked the rings from $Cr_2N$ and $\alpha$).

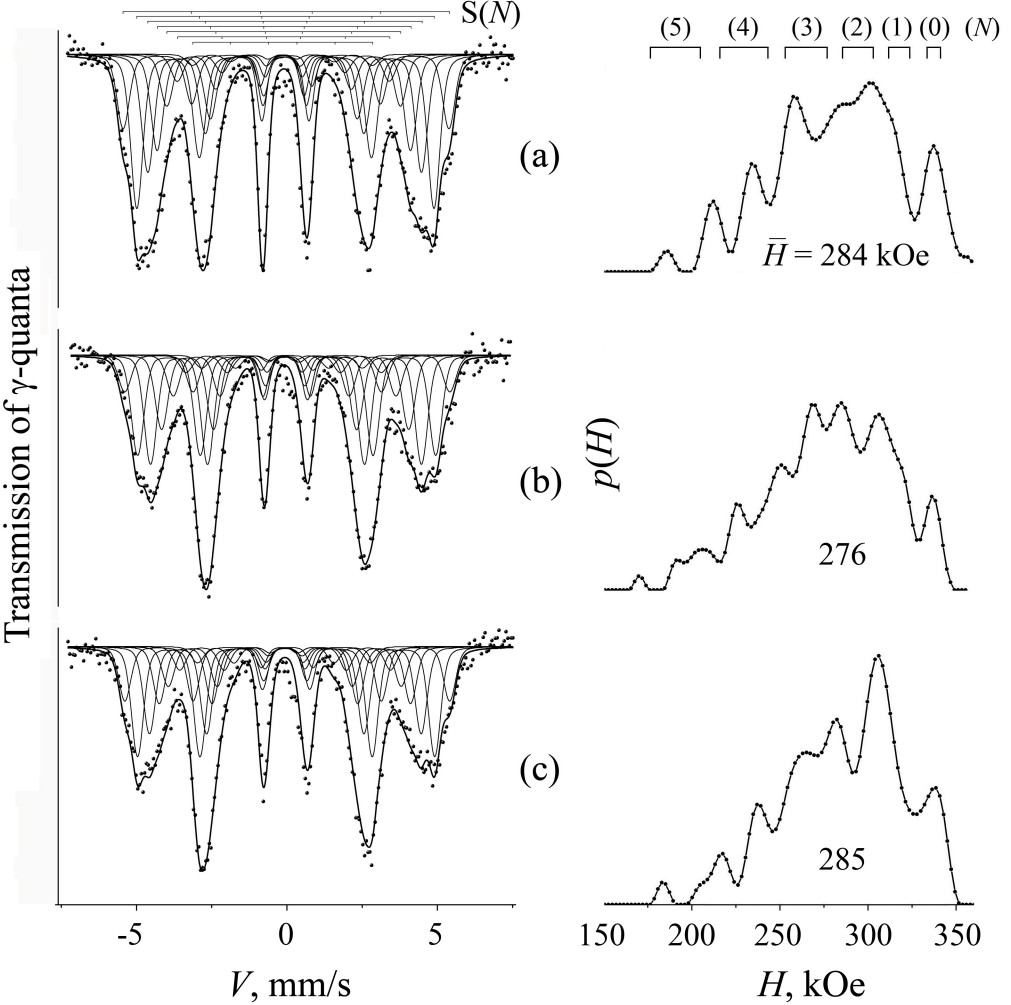

**Figure 8.** Spectra and the distribution $p(H)$ for ferrite of the steel $Fe_{71.2}Cr_{22.7}Mn_{1.3}N_{4.8}$. Treatment (state): (**a**) annealed at 923 K for 2.5 h (aged); (**b**) aged and HPT treated at 80 K; (**c**) aged and HPT treated at 573 K. $S(N)$—the subspectra-sextets with $N$ atoms of Cr in the surrounding of Fe.

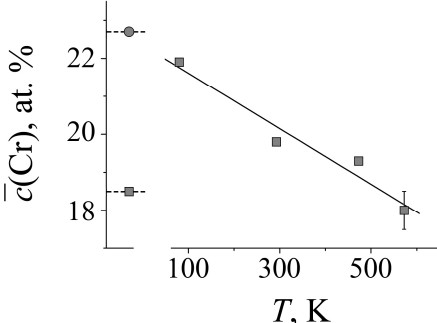

**Figure 9.** The change of the content of chromium, $\bar{c}(Cr)$, in the matrix of ferrite of the steel $Fe_{71.2}Cr_{22.7}Mn_{1.3}N_{4.8}$ aged at 923 K for 2.5 h in dependence of the temperature of HPT deformation. The composition of austenite of the as-received steel (i.e., water-quenched from 1473 K) is shown by a round symbol.

In the carbon austenitic steel $Fe_{64.7}Ni_{33.6}C_{1.7}$—upon megaplastic deformation—one can also observe active processes of dissolution of the graphite particles and disperse lamella of carbides [19,22], Figure 10. The translocation of carbon from particles into a metallic matrix and that in the reverse direction was analyzed based on the magnetic structure parameters and mean hyperfine magnetic field $\bar{H}$ of the austenite spectra. This is possible, since on alloying by carbon the FCC Fe-Ni alloys of invar composition sharply increase in $T_C$ and $\bar{H}$, as a consequence of suppressing antiferromagnetism [48,49]. The influence of deformation temperature was studied on the steel, water-quenched from 1423 K (as-received condition) and aged at 873 K for 1 h. Aging at 873 K, 1 h entails a decrease in the value of $\bar{H}$ from 257 to 240 kOe, which is a consequence of carbon leaving the matrix of austenite: the concentration of carbon thus decreases from 1.46 to 0.1 [50], see Figure 11a,b and Figure 12. «cold» (80, 298 K) deformation of aged steel leads to the opposite changes, namely, to an increase of $\bar{H}$ and to a decrease of the peak of antiferromagnetic component in the distribution $p(H)$, Figure 11c. Note that the observed changes in the hyperfine magnetic structure of a carbon-containing austenite are opposite to the changes caused by the destruction of the homogeneous short-range order in the binary alloy $Fe_{64.9}Ni_{35.1}$, see Figure 6a. The growth of $\bar{H}$ in the austenite of $Fe_{64.7}Ni_{33.6}C_{1.7}$ is explained by dissolving the graphite particles and carbides as a result of a dislocation-assisted transfer of carbon atoms from the particles into the matrix. As a result of the «cold» deformation of the steel $Fe_{64.7}Ni_{33.6}C_{1.7}$ aged at 80 K, in it (i) there forms an ultra-fine-grained misoriented structure and (ii) there takes place an intense fragmentation of lamella of the carbides $(FeNi)_3C$ with the formation of carbide particles of sufficiently smaller size (to 10 nm), Figure 10b,c. The character of atomic redistribution of carbon is cardinally changed in the experiments at «warm» (373–573 K) deformation of the as-received and aged steel, Figure 11d. When taking place, the intensive aging of austenite makes the carbon atoms actively leave the lattice of austenite, which proceeds at the expense of generation of a sufficient amount of deformation-induced point defects.

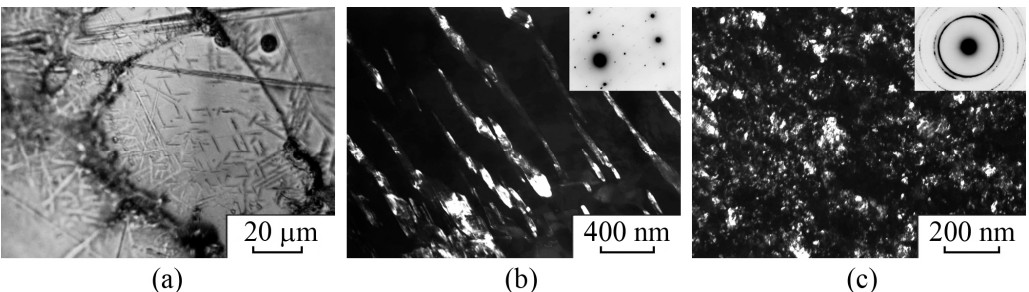

(a)    (b)    (c)

**Figure 10.** The structure of austenite of the steel $Fe_{64.7}Ni_{33.6}C_{1.7}$. Treatment (state): (**a,b**) annealed at 873 K for 1 h (aged); (**c**) aged and HPT treated at 80 K; (**a**) optical metallography; (**b**) dark-field image taken in the combined reflection $(211)_{(FeNi)3C}$; (**c**) dark-field image taken in the combined carbide–austenitic reflection $(111)_\gamma$.

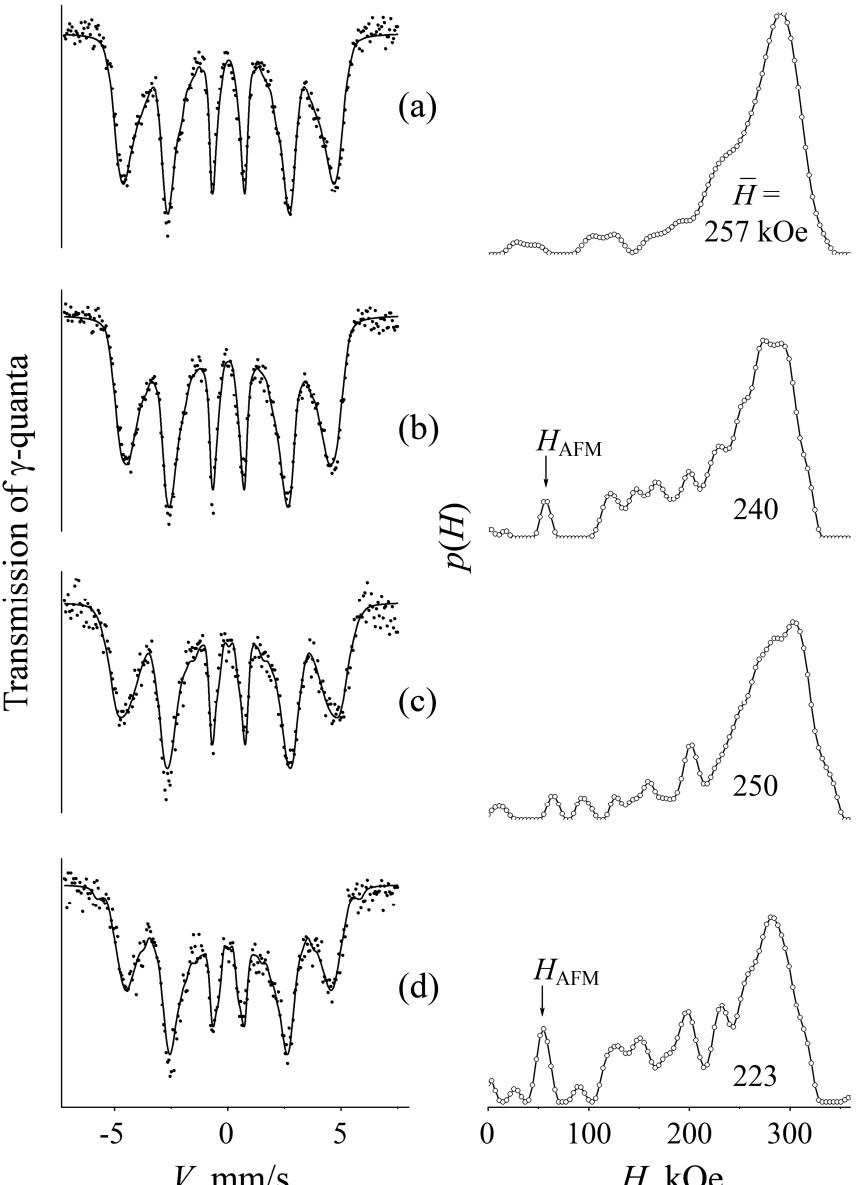

**Figure 11.** Spectra, $p(H)$, and $\overline{H}$ for austenite of the steel $Fe_{64.7}Ni_{33.6}C_{1.7}$. Treatment (state): (**a**) water-quenched from 1423 K (as-received); (**b**) annealed at 873 K for 1 h (aged); (**c**) aged and HPT treated at 298 K; (**d**) aged and HPT treated at 473 K. $H_{AFM}$ is the peak of antiferromagnetic component.

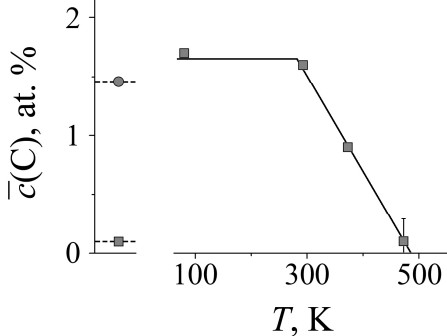

**Figure 12.** The dependence of the carbon content $\overline{c}(C)$ in the austenite matrix of the steel $Fe_{64.7}Ni_{33.6}C_{1.7}$ annealed (i.e., aged) at 873 K for 1 h on the temperature of HPT deformation. The round symbol shows the (concentration) composition of austenite of the as-received steel (i.e., water-quenched from 1423 K).

*3.2. The Regulation of the Chemical Composition of Dispersion-Hardened Steels at Megaplastic Deformation*

In the studied steels with the structure of austenite or ferrite, megaplastic deformation at 80 and 298 K leads to the dissolution of the products of decomposition (i.e., nitrides and carbides) located in the steel matrix and to the increase of the content of interstitial chemical elements (i.e., nitrogen and carbon) in solid solution. The amount of nitrogen and carbon in the austenite of the steels quenched from a high temperature is always smaller than in the steels after HPT mega-deformation at low temperatures. This is connected to an incomplete dissolution of the carbides and nitrides in the course of the process of heating for subsequent quenching and with a possible partial precipitation of interstitial phases during quenching [19,22,23]. The observed growth of chromium content from nitrides located in the steel matrix under deformation-induced dissolution when lowering the temperature of deformation from room to cryogenic in value, is explained by the weakening of the component of dynamic aging, Figure 9. Of interest is also another phenomenon, the anomalous acceleration of diffusion-controlled decomposition upon «warm» deformation (at the expense of intensive generation of deformation-induced point defects), in comparison with the case of thermal annealing at sufficiently higher temperatures. Thus, the degree of decomposition of carbon-containing austenite in conditions of HPT at 473 K over the course of 10 min (three revs with a rate of 0.3 rpm), surpasses the result of austenite decomposition observed after annealing at 873 K over the course of one hour. Such acceleration is observed also in the ferrite of the steel $Fe_{71.2}Cr_{22.7}Mn_{1.3}N_{4.8}$. In Section 2 the processes of anomalous acceleration of diffusion-controlled are demonstrated on the example of short-range clustering of binary Fe-Mn, Fe-Cr, and Fe-Ni alloys. Therefore, the reasons for the realization of acceleration of deformation-induced decomposition in steels can be the generation of a large quantity of vacancies as well as the growth of their mobility on rising the temperature of deformation. However, as in the case of irradiation by high-energy particles, the continuous generation of mobile point defects is the main condition for acceleration of decomposition. The continuity of deformation action is the cause of sustaining a high concentration of the interstitial atoms and vacancies, a necessary condition for realization of lattice diffusion at relatively low temperatures [4–6,43,44]. It is important to note that in the processes of the dynamic dissolution and formation of particles, a great role is played by active «excited» transcrystalline boundaries. The participation of transcrystalline boundaries in the processes of dynamic recrystallization at large plastic deformation is described in a number of papers and reviews [34,51,52].

The diffusion-controlled transformations induced by HPT are observed also by friction-providing action. Thus, in conditions of dry friction of sliding, in the surface layers of austenite steel $Fe_{56.9}Mn_{21.5}Cr_{18.6}N_3$, there takes place dissolution of the products of cellular decomposition, namely, $Cr_2N$ nitrides, which improves the characteristics of wear resistance of the steel [21]. Within the context of the obtained results on dissolving chromium nitrides mentioned above, the work [32] is of particular interest, where cases of HPT deformation and dry friction of sliding are compared. There it is shown that the HPT method simulates conditions of friction action at which, on contact of loading, a large deformation is realized in conditions of compression.

Thus, by changing the temperature of the large pressure-torsion or sliding-friction plastic deformation we have a means to regulate the ultra-fine-grained structure, the composition of the metallic matrix, as well as the quantity, and dispersity of the interstitial phases in the high-carbon and high-nitrogen steels.

## 4. Mechanical Alloying in Aging FCC Fe-Ni-Me (Me = Ti, Al, Si, Zr) Alloys

*4.1. The Effect of the Temperature and Rate of Megaplastic Deformation on the Structural Transformations in Austenitic Fe-Ni-Me Alloys. A Phenomenological Model of Mechanical Alloying*

It is known that in binary Fe-Ni alloys long-term annealing with subsequent irradiation by electrons leads to the formation of regions of short-range atomic order, Fe-Ni clusters, or ordered FeNi and $Ni_3Fe$ phases [37–39]. Doping of Fe-Ni alloys by Ti, Al, Si, and Zr favors the development of

another process, aging with the formation (in the initial stages) of (i) clusters and aggregates of atoms of nickel with Ti, Al, Si, and Zr [53], and then, with a large degree of decomposition, (ii) precipitations of ordered intermetallic phases of the type $Ni_3(Ti, Al, Si, Zr)$, Figure 13.

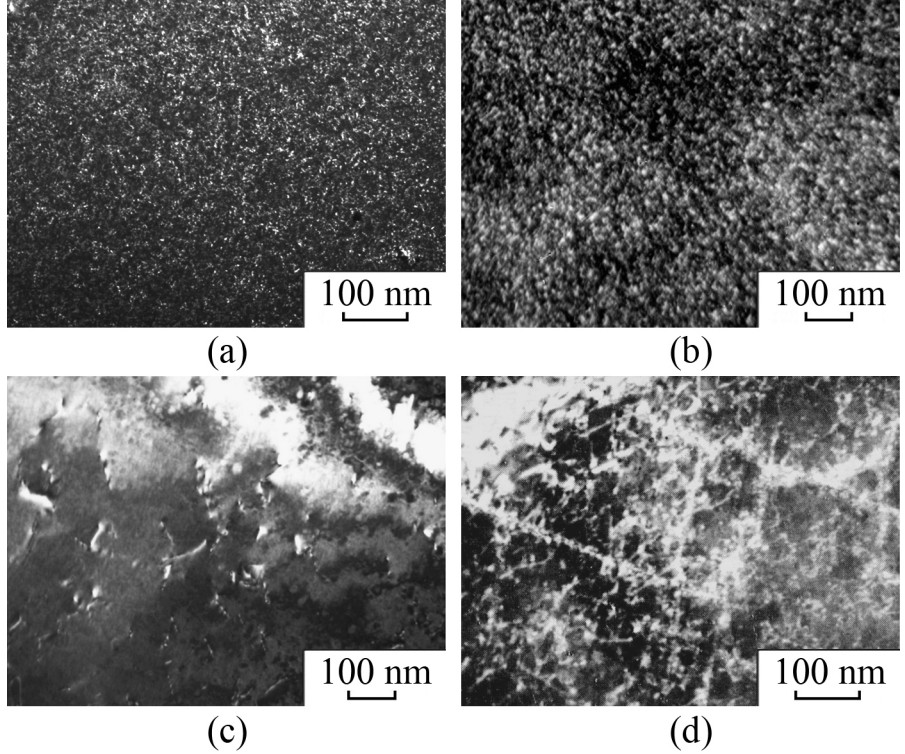

**Figure 13.** Dark-field images taken in the matrix reflection of the alloys after aging at 923 K for 3 h. (**a**) $Fe_{63.5}Ni_{33.4}Ti_{3.1}$, (**b**) $Fe_{55}Ni_{36}Al_9$, (**c**) $Fe_{62.1}Ni_{32.2}Si_{5.7}$, (**d**) $Fe_{62.9}Ni_{34.9}Zr_{2.2}$.

Fe-Ni-(Ti, Al, Si, Zr) alloys are not of a low-melting series; however, in these alloys an active atomic redistribution at the rolling to high reduction degrees has been revealed as a consequence of the deformation-induced dissolution of intermetallics in a metallic matrix. The activity of deformation-assisted dissolution of the intermetallic $\gamma'$ phase depends on the peculiarities of intermetallics and is realized at large critical deformations of $\varepsilon \geq 4$ [5]. Besides, on lowering the temperature of deformation from room to cryogenic in value, in the works [13–16] the acceleration of the dissolution of intermetallics was established. The quantitative estimates of the dissolution and precipitation of intermetallics were carried out based on the content of nickel in the FCC matrix of the alloys and on the strong dependence both of magnetic characteristics ($T_C$, magnetization, etc.) and of the mean hyperfine magnetic field $\overline{H}$ on the nickel concentration $\overline{c}(Ni)$ in the metallic matrix [26,48]. In Figure 14a–c one can see the spectra and distributions $p(H)$ for the alloy $Fe_{55}Ni_{36}Al_9$ after aging and HPT deformation at 80 and 537 K, and in the inset to the figure, the dependence of $\overline{H}$ on the nickel content in the Fe-Ni alloys [26]. It is obvious that the decrease of $\overline{H}$ in value after aging is stipulated by the change in nickel of its location from its (Ni) surrounding by iron in the matrix to the intermetallic $\gamma'$ ($Ni_3Al$) phase. Lowering the temperature of deformation from room-value to 80 K in the HPT experiment and rolling entails the growth of $\overline{H}$ in consequence of the dissolution of the $\gamma'$ phase and entering the austenite matrix by nickel. «Warm» deformation at 473–573 K leads to an opposite effect, the decrease of $\overline{H}$ for the alloy $Fe_{55}Ni_{36}Al_9$ water-quenched from 1373 K (as-received condition) and aged at 873 K for 3 h as a consequence of an additional amount of nickel exiting from the matrix, Figure 14d. The behavior of $\overline{c}(Ni)$ in dependence of the deformation temperature is shown in Figure 15. Thus, the change of temperature of the megaplastic deformation from «cold» (80–298 K) to «warm» (473–573 K) is accompanied by a change in the direction of development of the phase

transformation, from dissolution to precipitation of intermetallics. It is important to note that the activity of the processes of aging, as well as of dissolution, is essentially connected with the dynamic origin of the processes, namely, with the continuity of the generation of point defects, which take part in the process of atomic mass transfer. This is clearly seen when comparing results of dynamic aging at 573 K and thermal aging under annealing at 573 K, of the alloy preliminarily HPT deformed at room temperature (point *c* in Figure 15). The soaking duration at 573 K (50 min) in both cases is the same, but in HPT conditions the sample was aged for 1.5 min in the dynamic regime. As compared with the result of preliminary aging at 873 K for 3 h, the dynamic aging at 573 K for 1–3 min as high as three times, makes the effective content of nickel in the matrix smaller because of precipitation of the intermetallic $\gamma'$ phase. The structure of the alloy $Fe_{55}Ni_{36}Al_9$ after the deformation is characterized by the refinement of the matrix grain, Figure 16a, b.

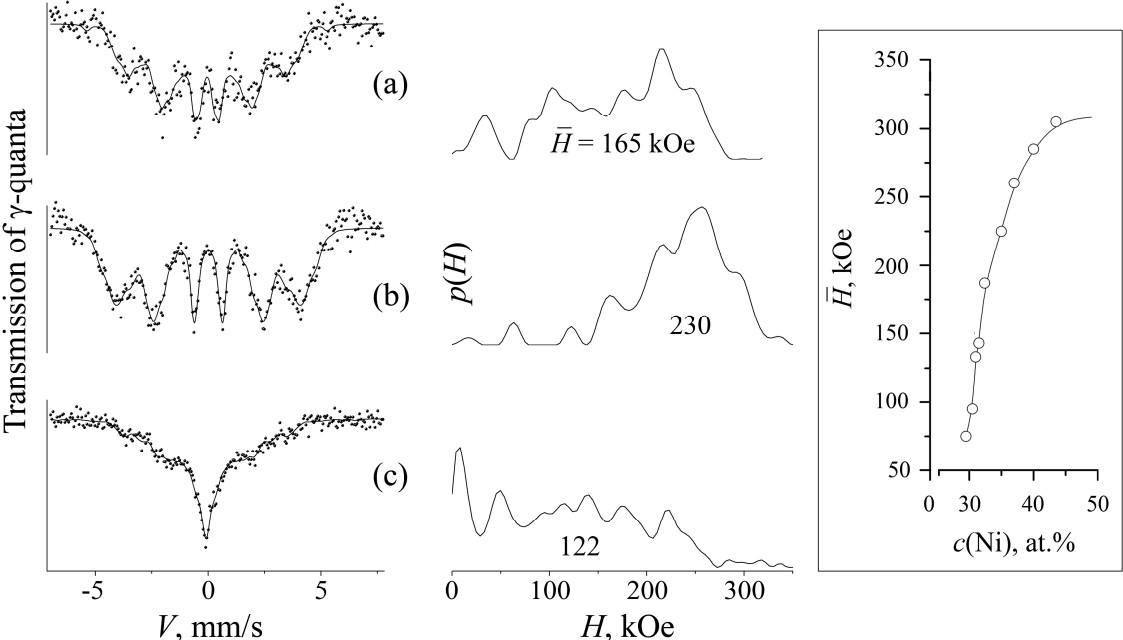

**Figure 14.** Spectra, the distributions $p(H)$, and magnitudes of $\overline{H}$ for the alloy $Fe_{55}Ni_{36}Al_9$. Treatment (state): (**a**) annealed at 873 K for 3 h (aged); (**b**) aged and HPT treated with $\varepsilon = 3.9$ (0.5 rev) at 80 K; (**c**) aged and HPT treated with $\varepsilon = 3.9$ (0.5 rev) at 573 K. In the inset we show the dependence $\overline{H} = f(c(Ni))$ for FCC binary Fe-Ni alloys borrowed from [46].

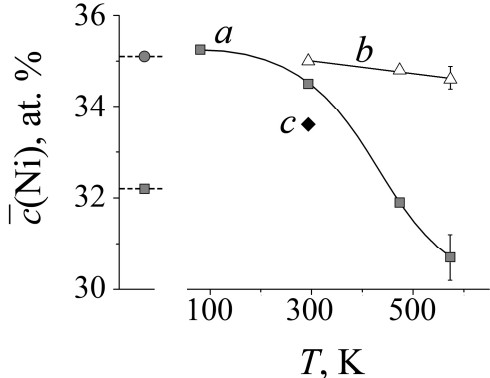

**Figure 15.** The effect of deformation temperature on the nickel concentration $\overline{c}(Ni)$. (**a**) in austenite of the alloy $Fe_{55}Ni_{36}Al_9$ aged at 873 K for 3 h; (**b**) in the binary alloy $Fe_{64.9}Ni_{35.1}$; (**c**) in austenite of the alloy $Fe_{55}Ni_{36}Al_9$ preliminarily aged at 873 K for 3 h after its HPT treatment with $\varepsilon = 4.8$ (1 rev.) and subsequent isothermal annealing at 573 K for 50 min. The round symbol corresponds to the value characteristic of the state obtained after quenching from 1373 K in water.

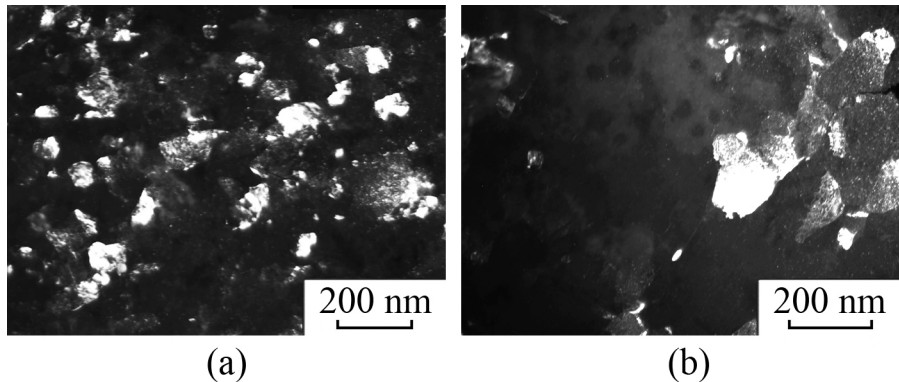

**Figure 16.** The structure of the alloy $Fe_{55}Ni_{36}Al_9$, and dark-field images taken in the reflection $(111)_\gamma$. Treatment (state): (**a**) annealed at 873 K for 3 h (aged) and HPT treated with $\varepsilon = 5.9$ (3 revs) at 298 K; (**b**) aged and HPT treated with $\varepsilon = 5.9$ (3 revs) at 573 K.

For analysis of the mechanism of dynamic aging, experiments on the effect of the rate of deformation are of great importance [13,14]. In the alloy $Fe_{55}Ni_{36}Al_9$ aged at 873 K for 3 h, it was established that the rise of the rate of deformation from $2.4 \times$ s$^{-1}$ (0.3 rpm) to $8 \times 10^{-2}$ s$^{-1}$ (1.0 rpm) entails the increase of nickel content in the matrix. The effect of the rate of deformation increases after HPT at 473 K ($\Delta\bar{c}(\mathrm{Ni}) \sim 1$ at %), see Figure 17. These data, as well as the data on the effect of the temperature of deformation on the direction of the evolving phase transformations, are a confirmation for the rise of the role of vacancy mechanism in the range of temperatures of megaplastic deformation under our investigation.

In the works [5,13,14], on the basement of experimental data on the kinetics of dissolution of intermetallics, we proposed a phenomenological model, with the base equation in the form:

$$\Delta c(\mathrm{Ni}) = K_1(\varepsilon - \varepsilon_{\mathrm{cr}}) + K_2 \cdot D \tag{2}$$

where $\varepsilon$ is the degree of true deformation; $\varepsilon_{\mathrm{cr}}$ is the critical deformation beyond which the process of dissolution of intermetallics begins; $D$ is the coefficient of diffusion; $K_1$ and $K_2$ are coefficients of proportionality. The first term in Equation (2) is weakly dependent on the temperature and describes the change of composition as a result of (i) cutting (while "passing-by") particles by moving dislocations and (ii) successive removal of atoms from particles into the matrix. Since the true deformation $\varepsilon$ is determined by the full number $q$ of dislocations that have already crossed the particle and by the average length $l$ and the (abs. value of) Burgers vector $b$, i.e., $\varepsilon \sim qlb$, then the first term in Equation (2) can be written in the form $\Delta c(\mathrm{Ni}) \sim K_1 qlb$ [5]. Together with the electron-microscopy data [54], this permitted us to draw a conclusion on the dislocation-related origin of the dissolution of intermetallic particles in the course of «cold» deformation. Theoretical estimates [29] have shown that lattice (-mediated) diffusion is feasible at low temperatures, if it happens to evolve in the stress field of moving dislocations due to migration of interstitial atoms (the activation energy of which amounts to 0.1–0.3 eV). The second term of Equation (2) depends on the temperature and rate of deformation, and its appearance is connected with the saturation of the structure by mobile vacancies. This term describes the process of dynamic aging, namely, its relaxation-component part of the megaplastic deformation-induced phase transformation.

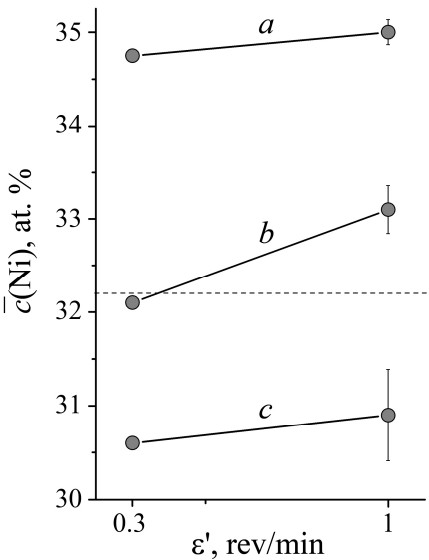

**Figure 17.** The dependence of the nickel content $\bar{c}(\text{Ni})$ in the matrix of the alloy $Fe_{55}Ni_{36}Al_9$ on the rate $\dot{\varepsilon}$ of its deformation at temperatures (**a**) 293 K, (**b**) 473 K, and (**c**) 573 K on the degree of deformation equal to $\varepsilon = 4.8$ (1 rev.).

*4.2. The Effect of the Activity of Chemical Elements on the Kinetics of Mechanical Alloying. Dual Nature of Dissolution of Intermetallic Particles (i) upon Deformation and (ii) in the Cascades of Atomic Displacements upon Irradiation by Fast Neutrons*

Experimental data on the effect of the diffusion mobility and activity of chemical elements on the kinetics of mechanosynthesis, confirm an idea on the dual nature of the mechanism of mechanical alloying, namely, as concurrent competitive evolvement of the deformation-induced dissolution and precipitation of intermetallics. For analysis of the structure–phase transitions upon megaplastic deformation, we conducted experiments on the intense irradiation effect of electrons and neutrons [14,17,19,20,53,55–57]. Such a comparison is explained by a specific feature common to both deformation and irradiation, namely, by a feature of saturation of the structure with mobile point defects. Earlier, the irradiation-induced dissolution of particles of the intermetallic γ′ phase was revealed on the Fe-Ni-(Ti, Al, Si, Zr) alloys, in the cascades of atomic displacements, upon irradiation of the alloys by fast neutrons [55–57]. The critical size of intermetallic particles was established, at which an active dissolution takes place and which is determined by its commensurability with the dimensions of the cascades of atomic displacements [55]. In the case of mechanical-induced dissolution, the kinetics of the process correlates with the dimensions of the particles and the probability of an encounter between dislocations and particles [27]. The general regularity of the effect of the deformation temperature and irradiation by fast neutrons on the processes of dissolution of intermetallics was established on the aged alloy $Fe_{63.5}Ni_{33.4}Ti_{3.1}$. It was shown [14] that the irradiation at 340 K of the alloy preliminarily aged at 923 K for 0.5 h, changes the magnitude of $\overline{H}$ in an opposite manner (based on the spectra), which testifies to the increase by 0.2 at % at 340 K, and to the decrease by 0.5 at % at 530 K, of the content of nickel in the matrix, Figure 18. Consequently, on raising the temperature of irradiation, the mobility of vacancies and their participation in irradiation-induced phase transformations grow.

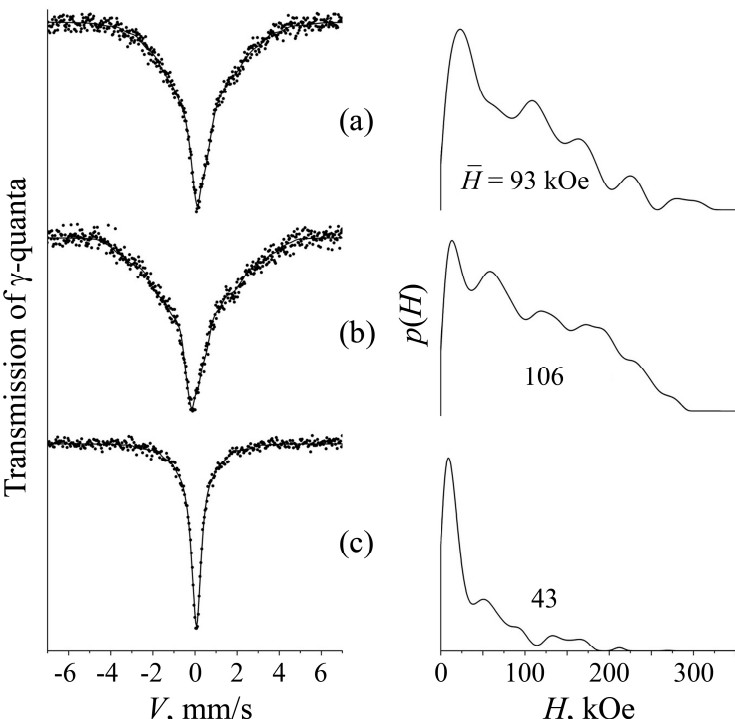

**Figure 18.** Spectra, the distributions $p(H)$, and magnitudes of $\overline{H}$ for the alloy $Fe_{63.5}Ni_{33.4}Ti_{3.1}$. Treatment (state): (**a**) annealed at 923 K for 0.5 h (aged); (**b**), (**c**) aged and irradiated by fast neutrons with a fluence of $10^{20}$ cm$^{-2}$ at 340 K and 530 K, respectively.

As confirmation of the common «dual» nature of the phase transitions induced both in the course of megaplastic deformation and in the cascades of atomic displacements, there are common regularities of the effect on these processes of the chemical activity and diffusion mobility of intermetallide-forming chemical elements (Ti, Al, Si, Zr) [14,17]. In particular, the substitution in FCC Fe-Ni-(Ti, Al, Si, Zr) alloys of the chemically active Al and Ti by the less active Si increases both the change of <$H$> and the relative volume of dissolution of the intermetallics $Ni_3Si$ in the aged alloy $Fe_{62.1}Ni_{32.2}Si_{5.7}$ under conditions of cascade-forming irradiation by fast neutrons [17], see Figure 19. The bar diagram in Figure 20 demonstrates a relative change in the content of nickel (Z) in the matrix of the alloys aged at 923 K for 3 h after HPT at 298 K and neutron irradiation at 340 K. For $Z = 1$ the change of nickel content was taken in the matrix of the aged alloys after cold HPT (at 298K). In the work [53] it was shown that silicon less actively than titanium and aluminum forms the intermetallic $\gamma'$ phase in Fe-Ni austenite under conditions of thermal annealing and irradiation by electrons. To estimate the contribution from the formation of intermetallics to the kinetics of phase transformations, we performed an irradiation by electrons at 420 K of the alloys, disordered as a result of their deformation [16]. As when irradiating Fe-Ni-(Ti, Al, Si) alloys by fast neutrons, as when deforming, there takes place dissolution and precipitation of intermetallics [14,17,57]. One has a right to suppose that the efficiency of dissolution of intermetallics in the alloy with silicon is a consequence of the weak activity of concurrent aging.

The alloy doped by zirconium, for instance, $Fe_{62.9}Ni_{34.9}Zr_{2.2}$, behaves in a specific way. When it is irradiated by neutrons, no dissolution of an intermetallic phase is observed. Atomic redistribution proceeds along the way of the ordering of Fe and Ni in the case analogous to that which is typical of the binary alloy $Fe_{64.9}Ni_{35.1}$, see Figure 21 and Section 2, Paragraph *2.1*. The reasons for the low efficiency of dissolution are the large sizes of $Ni_3Zr$ particles and the low diffusion-provided mobility of zirconium. However, the HPT deformation of the alloy at room temperature is accompanied by (i) dissolution of particles of intermetallics and (ii) formation of $\alpha$ martensite of deformation. In contrast to the experiment on irradiation by electrons, the observed phase transition is a consequence of the relaxation of elastic deformation along the way, characteristic of a phase transition of the shear type [17].

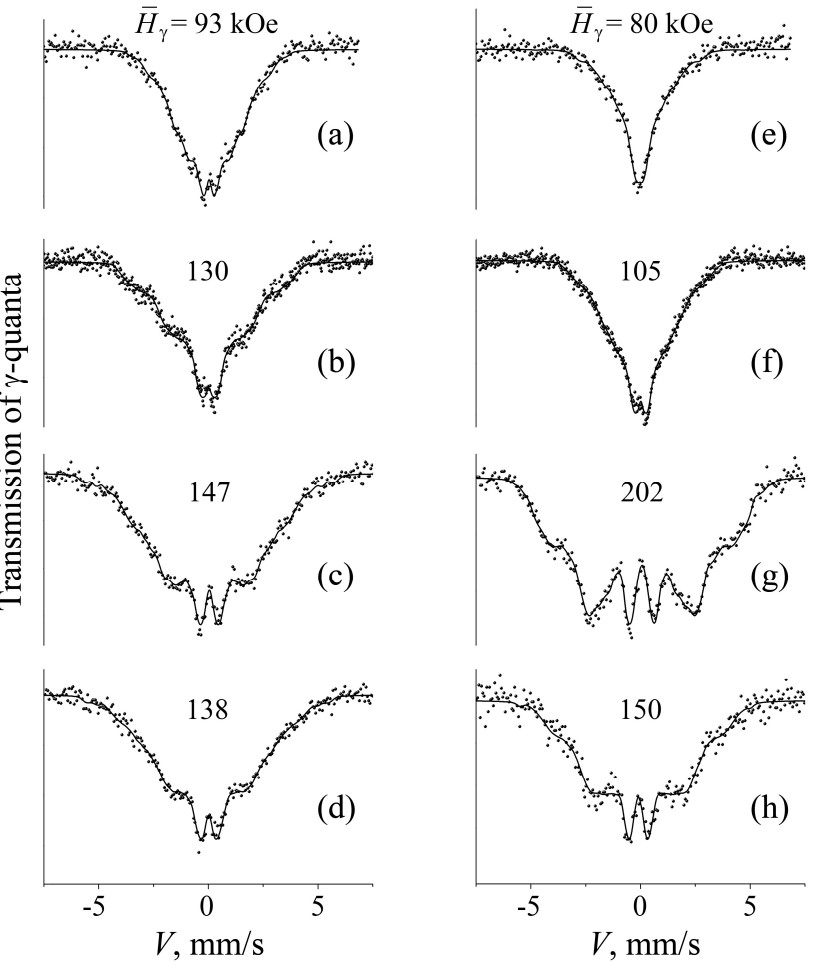

**Figure 19.** The spectra and magnitudes of $\overline{H}$ for the alloys (**a–d**) $Fe_{62.1}Ni_{32.2}Si_{5.7}$ and (**e–h**) $Fe_{63.5}Ni_{33.4}Ti_{3.1}$. Treatment (state): (**a,e**)–annealed at 923 K for 3 h (aged); (**b,f**)–aged and irradiated by fast neutrons with the fluence $F \sim 10^{20}$ cm$^{-2}$ at 340 K; (**c,g**)–aged and HPT treated with $\varepsilon \sim 7.2$ (5 revs) at 298 K; (**d,h**)–aged, HPT treated, and irradiated by electrons with $F \sim 10^{19}$ cm$^{-2}$ at 430 K.

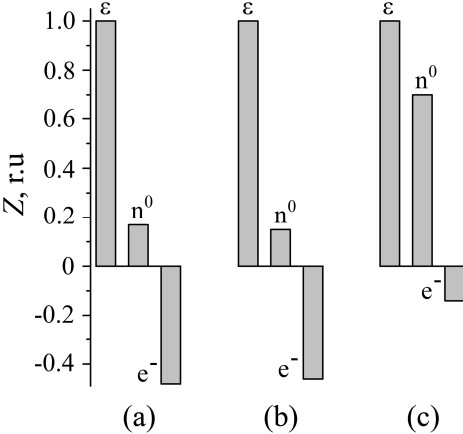

**Figure 20.** The bar diagrams of relative change of the content of nickel in the matrix (*Z*) of the alloys (**a**) $Fe_{63.5}Ni_{33.4}Ti_{3.1}$, (**b**) $Fe_{55}Ni_{36}Al_9$, and (**c**) $Fe_{62.1}Ni_{32.2}Si_{5.7}$, each aged at 923 K for 3 h upon various external actions. Treatment of the preliminarily disordered HPT alloys: $\varepsilon$—HPT (*n* = 5 revs) at 298 K; $n^0$—irradiation by neutrons at 340 K at the fluence $F \sim 10^{20}$ cm$^{-2}$; $e^-$—irradiation by electrons at 430 K with a fluence of $F \sim 10^{19}$ cm$^{-2}$.

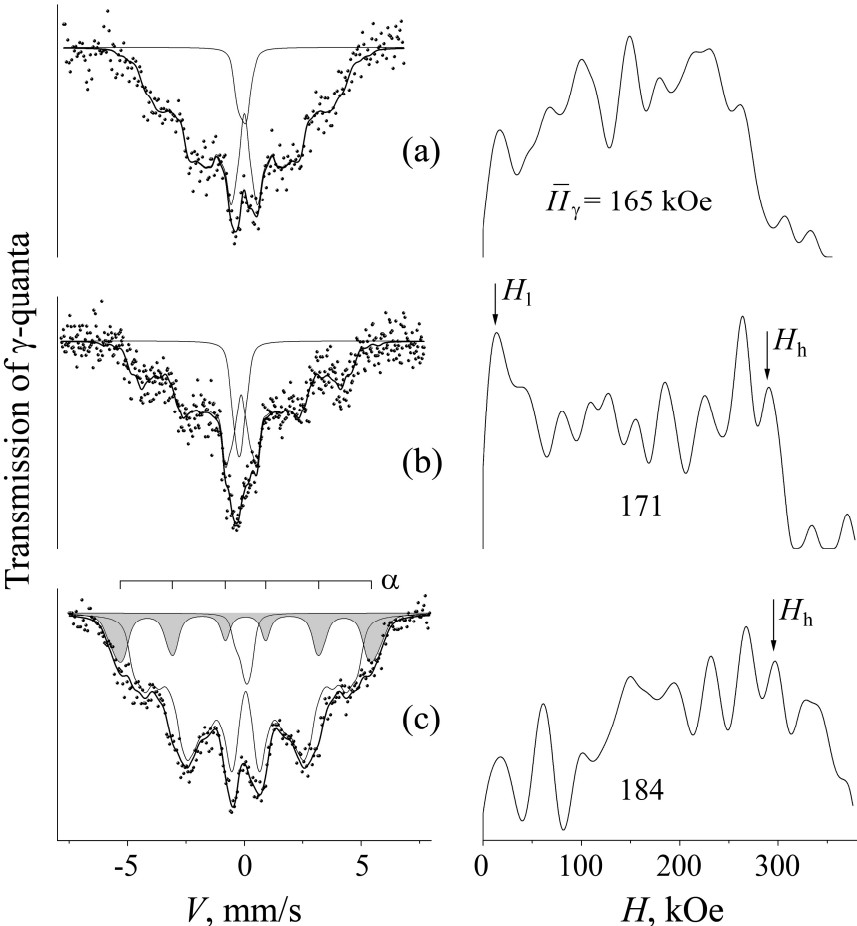

**Figure 21.** The spectra and distributions *p*(*H*) for the alloy Fe–35Ni–3Zr. Treatment (state): (**a**) annealed at 923 K for 3 h (aged); (**b**) aged and irradiated by fast neutrons with $F \sim 10^{20}$ cm$^{-2}$ at 340K; (**c**) aged and HPT treated with $\varepsilon \sim 7.2$ (5 rev) at 293 K. In (Figure 21) (**c**), the sextet of the $\alpha$ phase is marked in the spectrum. In the distributions *p*(*H*) the peaks $H_l$ and $H_h$, are pointed out that correspond to the $\gamma$ matrix depleted of, and enriched in, nickel.

## 5. The Modification of the Structure and Properties in Steel with Employment of Megaplastic Deformation (Conclusion)

The application of large plastic deformation in practice-important treatments (equal channel angular pressing, friction-providing action, BM milling, etc.) gives an opportunity to modify the structure and to propose ways of creating metallic materials with unique functional properties [1–5,8–12]. The attainment of the feasibility of controlling the conditions of large plastic deformation together with analyzing the structure and atomic redistribution at the nano-sized level is the principle condition for the implementation of these investigations. In the present work, this is achieved by (i) employment of the HPT method, by (ii) comparison with the case of irradiation effect, by (iii) using the capabilities of transmission Mössbauer spectroscopy, as well as of XRD methods of analysis.

Deformation-induced dissolution of the disperse phases permits the realization of schemes of nanostructurization, under which lies the deformation-induced formation of supersaturated solid solutions together with the initiation of extremely dispersed strengthening phases [9–12,21–23]. «cold» deformation in separate cases [5,9] can virtually present by itself a single means of obtaining the state of supersaturated solid solution in various steels and alloys.

At the same time, of great interest are the results on accelerating the processes of decomposition and precipitation of secondary strengthening disperse phases at «warm» megaplastic deformation. The change in the rate and temperature of deformation makes it possible to control and regulate the

ultra-fine-grained structure, as well as the amount, and the dispersity of the secondary, strengthening phases [13–23,32,33]. Secondary phases (intermetallides, carbides, nitrides, oxides, and others) hamper the growth of grains and provide conditions for the formation of a thermally stable structure. The employment in the capacity of oxygen carriers of the low-stable oxides ($Fe_2O_3$, $Fe_3O_4$, and others) or low-stable nitrides ($Fe_4N$, $CrN$, and others), which are easily dissolvable in the matrix of alloys at «cold» deformation under mechanical alloying, permits the development of novel advanced technologies for obtaining heat-resistant steels and alloys strengthened by special oxides ($Y_2O_3$, $TiO_2$, and others) [9,31,58,59] and nitrides ($Cr_2N$, $TiN$, $AlN$, and others) [5,10,11,14,33]. The surface layers modified by deformation—those of rubbing machine parts—can have increased wear resistance, which gives the opportunity to purposefully design new wear-resistant dispersion-hardenable materials with particles difficult to dissolve upon deformation. BM milling of a mixture of powders of nitrides ($CrN$, $MnN$, and others) makes it possible—in the iron matrixes and low-doped Fe-(Mn, Cr, Ni) alloys—to realize solid-phase mechanosynthesis of species of cost-saving alloyed nitrided Fe-(Mn, Cr, Ni)-N steels [10,11].

As can be seen from the above, an elucidation of the mechanism of the diffusion-assisted phase transformations in steels and alloys upon «cold» and «warm» intense megaplastic deformation opens new horizons in the creation of novel technological processes and novel materials with improved and advanced properties.

**Author Contributions:** Conceptualization, V.Sh. and V.S.; Methodology, V.Sh., V.S. and K.K.; Validation, V.Sh., V.S. and Y.U.; Formal Analysis, V.Sh., V.S. and K.K.; Investigation, V.Sh., V.S. and K.K.; Writing—Original Draft Preparation, V.Sh. and V.S.; Writing—Review and Editing, V.Sh. and Y.U.; Project Administration, V.Sh. and V.S.; Funding Acquisition, V.Sh.

**Funding:** This work was performed under the state assignment of the Federal Agency for Scientific Organizations of Russia (theme "Structure" No. AAAA-A18-118020190116-6) supported in part by the Russian Foundation for Basic Research (project No. 18-03-00216).

**Acknowledgments:** The authors are grateful to A.E. Zamatovsky, K.A. Lyashkov, N.V. Kataeva, L.G. Korshunov, V.P. Pilyugin, A.V. Makarov, and E.G. Volkova for their contribution in the conduction of experiments and for discussion of the obtained results.

**Conflicts of Interest:** The authors declare no conflict of interest.

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
