# Peer review of "Atomic Order and Submicrostructure in Iron Alloys at Megaplastic Deformation"

_metals, doi:10.3390/met8120995_

Round 1
Reviewer 1 Report
The manuscript mainly reports the results obtained by the authors in the field of the nanostructuring of alloys induced by plastic deformations. The topic is interesting and the experimental data are well organized and discussed, therefore I recommend it for publication after having properly addressed the following concerns.
1 - Of the 59 references, 12 are in Russian. This is not acceptable and the authors must strongly reduce the references in Russian.
2 - The english level is very poor. A deep revision is needful to eliminate typing and english mistakes.
- throughout the paper (lines 64,65,82,93,94,123,166,216,406,426,447,456,457,458,465,496,508, 526,530,533,551) I see words in parentheses with unclear meaning. The authors must fix the problem.
- line 93, change ""on the basement of" in "based on".
- 102, change "values" in "value".
- 120, change "impurity" in "impurities".
- 126, change "the lower degree of separation than that one" in "a degree of separation lower than that".
- 166,167 The phrase makes no sense. Please fix.
- 170, the title of the subsection is "Subsubsection"? Please fix.
- 172, change "in result of performing" in "resulting from".
- 202-205, please explain better.
- 215 change "by the fact of" in "as due to".
- 228 change "According XRD data" in "According to XRD data".
- 270, change "At an XRD pattern taken from" in "In the XRD pattern taken on".
- 290, change "In result" in "As a result"
- 291,292 change "in it there (i) forms an ultra-fine-grained misoriented structure and (ii) takes place an" in "(i) an ultra-fine-grained disoriented structure forms and (ii) it takes place an".
- 298,299 change "Taking place, the intense aging of austenite is accompanied by that the carbon atoms actively leave" in "The intense aging of austenite cause the carbon atoms to actively leave".
- 324, change "more higher temperatures in value." in "higher temperatures.".
- 410, the last part of the phrase makes no sense, please fix.
- 431, change "on the basement of" in "based on".
- 446-448 the phrase makes no sense, please fix.
- 495 change "When its irradiating" in "In fact, when it is irradiated".
- 521-524 the phrase "This,......analysis" makes no sense. Please fix.
- 526, change "in the basement of which (ones) lies the" due to".
Author Response
Dear Sir,
Thank you for your close peer-reviewing. It helped us exclude many mistakes.
We tried to meet the majority of your requirements.
The blue – means «accepted and revised».
The yellow – means «remained unaltered or slightly changed».
1 – Of the 59 references, 12 are in Russian. This is not acceptable and the authors
must strongly reduce the references in Russian. –
The majority of the references given in Russian are pioneering. Thereafter, in our next
(subsequent) publications, for instance, in ref.[5], we have presented references to these first
works. The authors consider it necessary – priority to be taken into account – to point out
references to the originals in Russian that were not translated to become further presented in
English. Here, the absence of [references to the] originals will – in our opinion – worsen the idea
of works that have formed the basis of the review.
We have interchanged only ref.[35] in favor of its version published in English.
2 – The english level is very poor. A deep revision is needful to eliminate typing and english mistakes. – see new revised version of the paper
3 - throughout the paper (lines
64,65,82,93,94,123,166,216,406,426,447,456,457,458,465,496,508, 526,530,533,551)
I see words in parentheses with unclear meaning. The authors must fix the problem. –
solved, [remained: Treatment (state) – annealed (aged)].
- line 93, change “”on the basement of” in “based on”.
- 102, change “values” in “value”.
- 120, change “impurity” in “impurities”.
- 126, change “the lower degree of separation than that one” in “a degree of separation
lower than that”.
- 166,167 The phrase makes no sense. Please fix.
- 170, the title of the subsection is “Subsubsection”? Please fix.
- 172, change “in result of performing” in “resulting from”.
- 202-205, please explain better. When analyzing the mechanism and kinetics of the atomic
mass transfer (AMT) induced by large-magnitude deformation, one should keep in mind its
combined character. In most of such cases, this makes one consider only the main contribution
into the AMT.
- 215 change “by the fact of” in “as due to”.
- 228 change “According XRD data” in “According to XRD data”.
- 270, change “At an XRD pattern taken from” in “In the XRD pattern taken on”.
- 290, change “In result” in “As a result”
- 291,292 change “in it there (i) forms an ultra-fine-grained misoriented structure and (ii)
takes place an” in “(i) an ultra-fine-grained disoriented structure forms and (ii) it takes
place an”. Not revised: [remained, as the construction (ii) is disputable and the term
disoriented, absent]
- 298,299 change “Taking place, the intense aging of austenite is accompanied by that
the carbon atoms actively leave” in “The intense aging of austenite cause the carbon
atoms to actively leave”. [revised but slightly]
- 324, change “more higher temperatures in value.” in “higher temperatures.”.
- 410, the last part of the phrase makes no sense, please fix.
- 431, change “on the basement of” in “based on”. [remained: as 2 “on”- too close]
- 446-448 the phrase makes no sense, please fix. Right You are!
- 495 change “When its irradiating” in “In fact, when it is irradiated”. The meaning lost.
[unchanged]
- 521-524 the phrase “This,......analysis” makes no sense. Please fix. Revised
- 526, change “in the basement of which (ones) lies the” due to”. [Unaltered]
Yours truly,
the authors

Reviewer 2 Report
Line 84: avoid the use of forms like: "we have considered…" and prefer the "third person …"
Fig.3: The (C) graph is reporting only three experimental condition/points, without linear correlation coefficient. It is suggested to insert at least another data otherwise it is suggested to discuss in the text the reason of only three points. Probably, the problem arises because the narrow window analysed 0-2 (F,cm-2) 10-18.
line 170: delete or re-edit...Probably a simple typing mistake.
line 514: Probably this chapter represents the "Conclusion" . In fact, in the paper there is any "Conclusion" chapter with a resume and an outline of your findings.
Author Response
Dear Sir,
We thank you for your critical remarks.
Trying to meet your requirements, we’ve made the following revision in the text of the paper:
line 84 – “we have considered” is substituted for by “one can find an authorized
consideration of”
Fig.3 , The revision of the caption: (a–b) The dependence (and (c) a tendency) ...
lines 166 –170 – deleted
line 514 – we add “(in conclusion)”
The yellow – means «the initial version».
The blue – means «the last version».
With best regards,
the authors

Reviewer 3 Report
Please see attached.

Author Response
Dear Sir,
We greatly appreciate your estimate of our work and your interest in the thematic
fundamentals (whose terminology is not wide-spread). The term «megaplastic deformation» is
introduced deliberately into our paper. This is not occasional. Let us here answer the questions
you‘ve set.
1. «Please, define the term “megaplastic deformation” or “megaplasticity” more universally»
The term of «megaplastic deformation» was introduced in the paper [7] (see below) by A.M.
Glezer with co-authors, with the purpose of formulating and giving the universal physical pattern of
the process at super-high plastic deformation of solid bodies. The physical meaning of this concept
consists in that, in consequence of the decline in the mobility of dislocations and a decrease in the
ability to the relaxation of the stored elastic energy along the way of structure fragmentation,
various process are possible to occur, namely, those of (i) dynamic recrystallization, (ii) stressassisted
diffusion, (iii) thermally activated diffusion, or (iv) amorphization. The border of
realization of the transition from the macro- to the mega «level» is determinated by switching-on of
the processes of changing the channels of relaxation along the way of the (i) atomic mass transfer,
(ii) recrystallization, (iii) formation of the nano structures. The term «megaplastic deformation»
does not correspond to the term «megaplasticity», though the concept of «plastic deformation» is
principally important.
2. «Is megaplasticity a material dependent concept?»
The border between the «mega-plastic» and the «macro-plastic» deformation is determined by
the properties of a structure, namely, by the (i) mobility of dislocations, (ii) energy of the generation
and migration of point defects, (iii) temperature of melting, and so on.
3. «What is the criterion on megaplastic deformation from the viewpoint of deformation
dynamics?»
The dependence of the border of the start of megaplastic deformation on the conditions (rate
and temperature) of the influence can be a criterion to the «dynamics» of deformation.
4. «Or can we argue megaplasticity from the microscopic phenomenological view point?»
Microscopic phenomenology can be a consequence of the general physical approach to a
phenomenon under consideration.
5. «I am especially interested in the constitutive relation in megaplasticity»
The authors do not pretend on the «justification» of the term of megaplastic deformation;
however, the revelation – in the experiments on intense plastic deformation – of the regularities
analogous to those obtained at irradiating by high-energy electrons, neutrons, in our opinion, really
permits using the criterion of Glezer. The term «intense plastic deformation» does not permit to
single out the dominant processes of relaxation of elastic energy along the way of the (i) dynamic
recrystallization, (ii) stress-assisted diffusion, (iii) thermally activated diffusion, and (iv)
amorphization.
[7] Glezer, A.M.; Metlov, L.S. Physics of megaplastic (severe) deformation in solids. Phys.
Sol. State. 2010, 52, 1162–1169, doi.org/10.1134/S1063783410060089
Finally, see an extended insert in the text.
Yours truly,
the authors
